# Lipid Variability and Risk of Cardiovascular Diseases and All-Cause Mortality: A Systematic Review and Meta-Analysis of Cohort Studies

**DOI:** 10.3390/nu14122450

**Published:** 2022-06-13

**Authors:** Shuting Li, Leying Hou, Siyu Zhu, Qian Yi, Wen Liu, Yang Zhao, Feitong Wu, Xue Li, An Pan, Peige Song

**Affiliations:** 1School of Public Health and Women’s Hospital, Zhejiang University School of Medicine, Zhejiang University, Hangzhou 310058, China; shutingli98@zju.edu.cn (S.L.); haddyhou@zju.edu.cn (L.H.); siyuzhu@zju.edu.cn (S.Z.); qianyizj@zju.edu.cn (Q.Y.); wenliu1219@gmail.com (W.L.); 2The George Institute for Global Health, University of New South Wales, Sydney, NSW 2050, Australia; wzhao@georgeinstitute.org.cn; 3The George Institute for Global Health, Peking University Health Science Center, Beijing 100600, China; 4Menzies Institute for Medical Research, University of Tasmania, Hobart, TAS 7000, Australia; feitong.wu@utas.edu.au; 5School of Public Health and the Second Affiliated Hospital, Zhejiang University School of Medicine, Zhejiang University, Hangzhou 310058, China; xueli157@zju.edu.cn; 6Ministry of Education Key Laboratory of Environment and Health, Department of Epidemiology and Biostatistics, School of Public Health, Tongji Medical College, Huazhong University of Science and Technology, Wuhan 430000, China; panan@hust.edu.cn

**Keywords:** lipid variability, cardiovascular diseases, mortality, meta-analysis

## Abstract

No consensus has yet been reached on the associations of lipid variability (LV) with cardiovascular diseases (CVDs) and all-cause mortality. We aimed to quantify the associations of different types and metrics of LV with CVDs and all-cause mortality. PubMed, Medline, and Embase databases were searched for eligible cohort studies published until 14 December 2021. Lipids included total cholesterol (TC), high-density lipoprotein cholesterol (HDL-C), low-density lipoprotein cholesterol (LDL-C), and triglycerides (TG). Metrics of variability included standard deviation (SD), coefficient of variation (CV), and variation independent of the mean (VIM). The primary outcomes were CVDs and all-cause mortality. Random-effects meta-analysis was used to generate a summary of the relative risks (SRRs). Sources of heterogeneity were explored by subgroup analysis and meta-regression. A total of 11 articles based on seven cohorts were included. Participants in the top quartile of TC variability had an increased risk of CVDs (vs. bottom quartile: TC-CV: SRR 1.29, 95% CI 1.15-1.45; TC-SD: 1.28, 1.15–1.43; TC-VIM: 1.26, 1.13–1.41, respectively) and all-cause mortality (vs. bottom quartile: TC-CV: 1.28, 1.15–1.42; TC-SD: 1.32, 1.22–1.44; TC-VIM: 1.32, 1.25–1.40, respectively). Participants in the top quartile of HDL-C variability had an increased risk of CVDs (vs. bottom quartile: HDL-C-CV: 1.11, 1.07–1.15; HDL-C-SD: 1.18, 1.02–1.38; HDL-C-VIM: 1.18, 1.09–1.27, respectively) and all-cause mortality (vs. bottom quartile: HDL-C-CV: 1.29, 1.27–1.31; HDL-C-SD: 1.24, 1.09–1.41; HDL-C-VIM: 1.25, 1.22–1.27, respectively). LDL-C variability was also associated with an increased risk of CVDs (for top vs. bottom quartile; LDL-C-SD: 1.09, 1.02–1.17; LDL-C-VIM: 1.16, 1.02–1.32, respectively) and all-cause mortality (for top vs. bottom quartile; LDL-C-CV: 1.19, 1.04–1.36; LDL-C-SD: 1.17, 1.09–1.26, respectively). The relationships of TG variability with the risk of CVDs and all-cause mortality were inconclusive across different metrics. The effects of SRR became stronger when analyses were restricted to studies that adjusted for lipid-lowering medication and unadjusted for mean lipid levels. These findings indicate that the measurement and surveillance of lipid variability might have important clinical implications for risk assessment of CVDs and all-cause mortality.

## 1. Introduction

Cardiovascular diseases (CVDs) are the leading cause of death worldwide [1]. In 2019, CVDs accounted for 38% of the 17 million premature deaths caused by noncommunicable diseases before the age of 70, putting an enormous strain on healthcare systems [2]. Dyslipidemia is a cluster of lipid aberrations, including high levels of total cholesterol (TC), low-density lipoprotein cholesterol (LDL-C), triglycerides (TG), and low levels of high-density lipoprotein cholesterol (HDL-C) [3]. Previous studies have established that dyslipidemia is a causal risk factor for CVDs, such as stroke [4,5,6], myocardial infarction (MI) [4,6], and coronary heart disease (CHD). Moreover, dyslipidemia is also associated with the risk of mortality [7,8]. Therefore, control of lipid levels has been regarded as a critical measure for preventing CVDs and mortality [9,10].

A growing body of epidemiologic evidence suggested that individuals’ lipid patterns over time could be governed by two dimensions: the mean level and intraindividual variability [11]. Lipid variability (LV) refers to the fluctuations in various types of lipids over time. Variability is usually assessed by the standard deviation (SD), coefficient of variance (CV), and variation independent of the mean (VIM) [11]. However, there is no unified definition of LV [11]. Recently, high variability in different types and metrics of lipids has been proposed as an additional indicator for risk of CVDs, as well as mortality [6,12,13,14,15]. A post hoc analysis from the Treating to New Target trial suggested that LDL-C variability, independent of mean LDL-C levels, was a powerful risk factor for cardiovascular events in patients with stable coronary artery disease [12]. Another study involving 3,656,648 participants from the Korean National Health Insurance System found that TC variability was a substantial risk factor for all-cause mortality, MI, and stroke, independent of mean TC levels and the use of lipid-lowering medication [13]. Nevertheless, a null association of TC and TG variability with stroke was revealed in a prospective cohort study based on a Chinese population, independent of mean lipid level and the use of lipid-lowering medication [16]. Moreover, no consensus has been achieved on the associations of LV with CVDs and all-cause mortality.

By far, the associations of LV with CVDs and all-cause mortality have not been synthesized via meta-analysis. To fill this gap in knowledge, we conducted a systematic review and meta-analysis of cohort studies to assess and quantify the associations of different LV metrics with CVDs and all-cause mortality. A series of subgroup analyses and meta-regression were further conducted to evaluate sources of heterogeneity and robustness of the results across subgroups.

## 2. Materials and Methods

### 2.1. Search Strategy and Study Selection

This study followed the Preferred Reporting Items for Systematic reviews and Meta-Analyses (PRISMA) guidelines [17,18]. The protocol was registered with the International Prospective Register of Systematic Reviews online protocol (PROSPERO, CRD 42021286042).

From database inception to 14 December 2021, we conducted a comprehensive English literature search in PubMed, Medline, and Embase to identify cohort studies that assessed the associations of LV with CVDs and mortality. Further details regarding the search strategy are provided in Appendix A.

The title and abstract of retrieved records were screened, followed by a full-text review by two independent reviewers (SL and LH). Eligible articles were included based on predetermined inclusion criteria: (i) cohort study; (ii) the exposure of interest was different types and metrics of LV (Appendix A); (iii) the outcomes were CVDs (including MI, CHD, heart failure (HF), stroke, atrial fibrillation (AF), and peripheral vascular disease [19]) or mortality (including all-cause mortality and cause-specific mortality); and (iv) effect sizes, such as hazard ratios (HRs) or relative risks (RRs) with the corresponding 95% CI or standard error, were reported. We excluded in vitro studies, animal studies, randomized-controlled trials, cross-sectional studies, and non-original studies (i.e., reviews, case reports, and protocols). Studies conducted on participants with prior CVDs or dyslipidemia-related disease (e.g., patients with ST-segment elevation myocardial infarction (STEMI), patients with non-obstructive coronary artery disease (CAD), patients who underwent percutaneous coronary intervention (PCI) or patients with familial hypercholesterolemia (FH)) and individuals who required emergency hospitalization, were also excluded. If there were multiple publications from the same study, the one with the most recent or comprehensive results was kept.

### 2.2. Data Extraction and Quality Assessment

Two reviewers (SL and LH) independently extracted data from the included articles and assessed the quality of these articles, followed by a cross-check for consistency. Discrepancies were resolved through discussion or consultations with the senior investigator (PS).

For each included article, basic characteristics were extracted as follows: authors, publication year, the country and setting where the study was based, cohort, study period, mean/median follow-up time, participant characteristics (i.e., number, mean age or age range, percentage of females), types of lipid (i.e., TC, HDL-C, LDL-C, and TG), variability metrics (i.e., CV, SD, VIM, average real variability (ARV), average successive variability (ASV), root mean square error (RMSE), SD of the residuals (SDR)), CVDs or mortality ascertainment, comparison level, and covariates included in the adjusted models. The adjusted RR/HR with the corresponding 95% CI were also extracted from each study. If estimates were available from multiple multivariable-adjusted models, we only extracted the most fully adjusted one. If a study simultaneously reported several estimates based on different types of lipid or metrics of variability, we extracted all estimates.

Due to the limited numbers of studies on ASV, ARV, RMES, and SDR for each type of lipid, we classified these variability metrics into the group of “Others.” For example, TC-ARV, TC-ASV, TC-RMSE, and TC-SDR were considered TC-Other. Given that different cohorts reported effect sizes in HR or RR, we treated HR as RR to ensure consistency [20]. Furthermore, we performed a standardized transformation from per SD increment RRs to top vs. bottom quartile RRs using the method provided by Chêne and Thompson [21]. In brief, after log transformation, comparison of the top vs. bottom quartile corresponds to 2.54 times the log RR of an SD increase.

Quality assessment was performed based on the Newcastle–Ottawa quality assessment scale (NOS) for cohort studies [22]. Up to a maximum score of nine stars, an article is evaluated based on selection (four items, one star each), comparability (one item, up to two stars), and exposure/outcome (three items, one star each). We assigned scores of included studies according to study quality criteria: (a) good quality if NOS ranked ≥ seven stars; (b) fair quality if NOS ranked four to six stars; and (c) poor quality if NOS ranked ≤ four stars. Considering that CVDs are a group of chronic noncommunicable diseases, the follow-up period was regarded as adequate during the quality assessment if its mean duration was at least 5 years. More details on quality assessment are shown in Appendix A.

### 2.3. Data Analyses

For studies that reported effect sizes for males and females separately, an overall RR for each study was generated using a fixed-effects meta-analysis. Any effect sizes stratified by different outcomes were treated as separate data points. The summary RRs (SRRs) (for top vs. bottom quartile) were generated using inverse-variance weighted random-effects meta-analysis and illustrated with forest plots. Furthermore, subgroup analyses and meta-regression were performed for studies exploring the associations between different types and metrics of LV with CVDs and all-cause mortality (number of data points ≥3) to evaluate sources of heterogeneity. The variables used for the subgroup meta-analysis included: subtypes of CVDs (MI, stroke, AF, and HF), gender (male or female), whether adjusted for mean lipid level or not, whether adjusted for lipid-lowering medication or not, whether adjusted for hypertension or not, whether adjusted for diabetes or not, whether adjusted for body mass index (BMI) or not, and whether adjusted for smoking or not. We also performed a sensitivity analysis that excluded the studies conducted on participants with hypertension and diabetes to assess the stability of the results.

Between-study heterogeneity was assessed with Cochran’s Q test and the I^2^ statistic. In Cochran’s Q test, significant heterogeneity was defined as a *p*-value of less than 0.10. The degree of heterogeneity across studies was considered high, moderate, and low by I^2^ cut-off values of 25% and 75% [23]. Publication bias was evaluated by inspecting funnel plots. All analyses were performed with STATA version 16.0 (StataCorp, College Station, TX, USA). We deemed statistical significance at *p* < 0.05, and all *p*-values were two-tailed.

## 3. Results

### 3.1. Identification and Selection of Studies

A literature search identified a total of 4673 records, of which 2842 were considered for screening of title and abstracts after removing the duplicates. After detailed screening against selection criteria, a total of 11 articles based on seven cohorts were finally included in analyses (Figure 1) [6,14,15,24,25,26,27,28,29,30,31].

### 3.2. Study Characteristics

The detailed characteristics of the 11 included articles in this meta-analysis are presented in Table 1. Of 11 articles, 4 articles were from South Korea, 5 were from China, and the other 2 were from America. The number of participants in each study ranged from 1473 to 5433098, and median follow-up years ranged from 4.2 years to 8.3 years. Nine included articles were of good quality (≥7 stars), and two were of fair quality (4 ≤ stars ≤ 6) (Appendix A).

### 3.3. Lipid Variability and Cardiovascular Diseases

All six cohort studies from 10 articles were included in the meta-analysis of LV and CVDs.

Compared with people in the top vs. bottom quartile of TC variability, TC-CV, TC-SD, TC-VIM, and TC-Other were associated with a 29%, 28%, 26%, and 27% higher risk of CVDs, respectively (TC-CV: SRR 1.29, 95% CI 1.15–1.45; TC-SD: 1.28, 1.15–1.43; TC-VIM: 1.26, 1.13–1.41; TC-Other: 1.27, 1.03–1.56, respectively; Figure 2 and Appendix A). Significant heterogeneity was observed across studies (I^2^_TC-CV_ = 92.4%, I^2^_TC-SD_ = 90.7%, I^2^_TC-VIM_ = 91.1%, and I^2^_TC-Other_ = 86.4%, all *p* < 0.001). HDL-C variability was also associated with an increased risk of CVDs (for top vs. bottom quartile; HDL-C-CV: 1.11, 1.07–1.15, I^2^ = 67.2%, *p* = 0.009; HDL-C-SD: 1.18, 1.02–1.38, I^2^ = 60.1%, *p* = 0.057; HDL-C-VIM: 1.18, 1.09–1.27, I^2^ = 91.6%, *p* < 0.001; HDL-C-Other: 1.07, 1.05–1.09, I^2^ = 0.0%, *p* = 0.501, respectively; Figure 2 and Appendix A). Compared with people in the bottom quartile of LDL-C-SD and LDL-C-VIM, those in the top quartile had an increased risk of CVDs (LDL-C-SD: 1.09, 1.02–1.17, I^2^ = 71.9%, *p* = 0.007; LDL-C-VIM: 1.16, 1.02–1.32, I^2^ = 49.7%, *p* = 0.114, respectively). The associations of LDL-C-CV and LDL-C-Other with CVDs were not significant (Figure 2 and Appendix A). Compared with people in the bottom quartile of TG-SD, those in the top quartile had a 5% increased risk of CVDs (TG-SD: 1.05, 1.02–1.09, I^2^ = 0.0%, *p* = 0.431). TG-CV, TG-VIM and other metrics of TG variability were not significant with CVDs (Figure 2 and Appendix A).

Sensitivity analyses excluding articles conducted on patients with hypertension or diabetes showed similar results for TC variability, HDL-C-CV, HDL-C-VIM, LDL-C-VIM, and TG variability with CVDs. The associations of HDL-C-SD, LDL-CV, and LDL-SD with CVDs were inconclusive (for top vs. bottom quartile; HDL-C-SD: 1.17, 1.00–1.38; LDL-C-CV: 1.08, 1.00–1.17; LDL-C-SD: 1.04, 1.00–1.08, respectively; Appendix A).

### 3.4. Lipid Variability and All-Cause Mortality

Six articles from four cohort studies were included in the meta-analysis of LV and all-cause mortality.

Compared with people in the bottom quartile of TC variability, those in the top quartile had an increased risk of all-cause mortality (TC-CV: 1.28, 1.15–1.42, I^2^ = 67.0%, *p* = 0.048; TC-SD: 1.32, 1.22–1.44, I^2^ = 51.0%, *p* = 0.130; TC-VIM: 1.32, 1.25–1.40, I^2^ = 18.4%, *p* = 0.294; TC-Other: 1.30, 1.21–1.40, I^2^ = 1.1%, *p* = 0.364, respectively; Figure 3 and Appendix A). Compared with people in the bottom quartile of HDL-C-CV, HDL-C-VIM and HDL-C-Other, those in the top quartile had an increased risk of all-cause mortality (HDL-C-CV: 1.29, 1.27–1.31, I^2^ = 0.0%, *p* = 0.343; HDL-C-VIM: 1.25, 1.22–1.27, I^2^ = 0.0%, *p* = 0.586; HDL-C-Other: 1.25, 1.23–1.27, I^2^ = 0.0%, *p* = 0.700; respectively; Figure 3 and Appendix A). However, only the study by Liu et al. [28] reported that HDL-C-SD was associated with a 24.0% higher risk of all-cause mortality (for top vs. bottom quartile; 1.24, 1.09–1.41). Compared with people in the bottom quartile of LDL-C-SD, those in the top quartile had a 17% increased risk of all-cause mortality (1.17, 1.09–1.26, I^2^ = 43.3%, *p* = 0.184). Compared with people in the bottom quartile of TG-SD, those in the top quartile had a 11% increased risk of all-cause mortality (1.11, 1.03–1.19, I^2^ = 0.0%, *p* = 0.605). More details about other metrics of LDL-C variability and TG variability with all-cause mortality are in Figure 3.

After excluding articles conducted among patients with hypertension or diabetes, only one study [28] reported insignificant associations of LDL-C-SD and TG-SD with the risk of all-cause mortality (Appendix A). Other metrics of LV with all-cause mortality were unchanged.

### 3.5. Subgroup Analysis and Publication Bias

The results of subgroup analyses and meta-regression are presented in Table 2 and Appendix A. A stronger association between TC variability and CVDs was found in a study [31] that did not adjust for mean lipid level compared with studies that did adjust (TC-CV: adjusted: SRR 1.25, 95% CI 1.12–1.40; unadjusted: 3.83, 2.03–7.25; TC-SD: adjusted: 1.24, 1.13–1.37; unadjusted: 4.43, 2.29–8.56; TC-VIM: adjusted: 1.23, 1.11–1.36; unadjusted: 3.87, 2.04–7.32; all *p*-values for subgroup differences <0.05). Compared with studies without adjustment for lipid-lowering medication, studies that adjusted found a stronger association between TC variability and CVDs (TC-CV: adjusted: 1.43, 1.17–1.75; unadjusted: 1.13, 1.06–1.21; TC-SD: adjusted: 1.52, 1.23–1.86; unadjusted: 1.13, 1.07–1.21; TC-VIM: adjusted: 1.49, 1.23–1.81; unadjusted: 1.13, 1.07–1.21; all *p*-values for subgroup differences < 0.001; Table 2 and Appendix A). Similar findings were found for subgroup analysis and meta-regression of the relationship between HDL-C variability, LDL-C variability, and CVDs (Appendix A). There was no heterogeneity in TG variability with CVDs in the subgroup analyses and meta-regression (Appendix A).

In subgroup analyses of TC variability with all-cause mortality, there were positive associations in all subgroups (Appendix A). In addition, a stronger association between TC-CV and all-cause mortality was found in studies that adjusted for lipid-lowering medication compared with unadjusted studies (TC-CV: adjusted: 1.30, 1.14–1.49; unadjusted: 1.21, 1.05–1.40; *p*-values for subgroup differences <0.05).

The funnel plots suggested a slight publication bias in the different LV metrics with CVDs and all-cause mortality (Appendix A).

## 4. Discussion

Lipid variability is increasingly proposed as a predictor for CVDs and all-cause mortality risk [6,12,13,14,15]. To the best of our knowledge, this study is the first meta-analysis to quantify the associations of different types and metrics of LV with the risk of CVDs and all-cause mortality in cohort studies. Findings from the present meta-analysis indicate that TC-CV, TC-SD, TC-VIM, HDL-C-CV, HDL-C-SD, HDL-C-VIM, and LDL-C-CV were positively associated with CVDs and all-cause mortality. However, the associations of TG variability with CVDs and all-cause mortality were inconclusive due to limited numbers of studies. These findings suggest that LV, especially the variability in TC and HDL-C, irrespective of the measurement used, were positively associated with CVDs and all-cause mortality and might play a future role in clinical risk assessment.

A review from Simpson et al. illustrated that visit-to-visit variability in lipoprotein (e.g., TC, HDL-C, LDL-C, non-HDL-C, and apolipoprotein B) and TG was found to be associated with CVD outcomes and all-cause mortality, independent of their mean absolute levels, each other, and their traditional risk factors [11]. Another review published in 2020 expanded adverse health outcomes (e.g., diabetes, end-stage renal disease, dementia) and added the newest studies related to LV and CVDs [32]. However, both reviews only provided an overview of relevant studies but did not quantify the association of LV with adverse health outcomes via meta-analysis. Our findings extend previous observations by focusing on cohort studies and quantifying the relationship of LV with CVDs and all-cause mortality.

In our meta-analysis, TC, HDL-C, and LDL-C variabilities were significantly associated with the risk of CVDs and all-cause mortality. Despite a growing body of evidence from epidemiological studies [30,33,34], the pathophysiological mechanisms for such associations remain unclear, and several plausible explanations have been suggested to support our findings. First, high LV may increase the fluctuation of atherosclerotic plaque components, causing repeated cholesterol crystallization and dissolution inside the confined area of plaques, affecting plaque stability, leading to plaque rupture, and eventually increasing the risk of CVD-related events [16,32]. Vedre et al. found that when cholesterol crystallizes and changes into a solid from a liquid, it forms sharp-tipped crystals that increase up to 45% in volume, damaging the plaque membrane mechanically [35]. Similarly, a rabbit experiment also revealed that intermittent hyperlipidemia induces experimental atherosclerosis more efficiently than constant hyperlipidemia [36]. Second, changes in cholesterol can also lead to endothelial dysfunction, oxidative stress, and inflammation, all of which are essential pathophysiological components of many diseases caused by metabolic dysfunction [37]. They may act as mediators for atherosclerosis, further inducing CVDs and even death [32,37]. Furthermore, confounding factors may explain the association of LV with adverse events. These include poor medication compliance and self-management, multimorbidity, certain medications, poor quality of life and lack of support, and infections [32,37]. Given the hypotheses mentioned above, it is not unexpected that our study found such positive associations between LV with CVDs and mortality. Previous studies also have found that LDL-C variability may have a greater impact on the occurrence of adverse health events in the group of patients with prior CVDs who are more prone to lipid fluctuations [13,38,39]. Nevertheless, the capacity of the limited number of studies with different exposure–outcome combinations may not allow us to draw robust conclusions regarding the insignificance of LDL-C variability and TG variability with CVDs and all-cause mortality of the general population in the sensitivity analysis. Thus, further studies to investigate the relationships of LDL-C and TG variability with CVDs and mortality risks are needed, as are further studies to examine the mechanisms by which higher cholesterol variability relates to CVDs development and whether a reduction in cholesterol variability can lower CVDs risk are required. We call for more researchers from different countries and regions to focus on different types and metrics of LV and adverse health outcomes.

Our study found that adjustment for lipid-lowering medication or not and adjustment for mean lipid level or not might be sources of heterogeneity between studies of LV with CVDs. There were some heterogeneities between studies that adjusted for lipid-lowering medication and studies without such adjustment for TC variability, HDL-C variability, and LDL-C variability with CVDs. It is likely because the beneficial effects of using lipid-lowering agents mitigated the impact of the high variability of TC, HDL-C, and LDL-C on CVDs [14]. Moreover, there were some heterogeneities between studies that adjusted for mean lipid levels and studies without such adjustment for TC and LDL-C variabilities and CVDs. It can be indicated that mean lipid level might exaggerate the role of LV in adverse health outcomes. These findings suggest that lipid-lowering medication and mean lipid level might be significant confounding factors in the relationship between LV and CVDs and highlight the need to adjust for these factors in future epidemiological studies.

There are several strengths to our study. First, we utilized a comprehensive literature search strategy across multiple bibliographic databases and a rigorous extraction process. Second, we included a substantial sample size of over five million participants from eligible cohort studies, which could ensure the accuracy of the results. Third, we used comprehensive lipid variability metrics to explore their relationships with CVDs and all-cause mortality, which enriches the previous studies and may help guide the subsequent development of standardized LV metrics. We also performed a sensitivity analysis that excluded the studies conducted on participants with hypertension and diabetes to assess the robustness of the results. The sensitivity analysis showed good robustness of the results. Moreover, we conducted between-study subgroup and meta-regression analyses to evaluate whether effect sizes of LV differ across characteristics of the studies and populations and to explore the potential sources of heterogeneity.

However, several limitations should be considered. First, due to the limited numbers of available studies, the LV effects on the ARV, ASV, RMSE, and SDR metrics could not be taken into account separately. Similarly, firm conclusions could not be drawn regarding the limited information on different types and metrics of LV with subtypes of CVDs (e.g., MI, CHD, HF, stroke) and subtypes of mortality (e.g., CVD mortality, cancer mortality). Second, given that different cohorts reported effect sizes in HR or RR, we treated HR as RR to ensure consistency. Third, there was high heterogeneity between studies in several analyses, but we conducted subgroup meta-analyses and meta-regression by common study characteristics and population characteristics to explore the sources of heterogeneity. There were some heterogeneities between studies that adjusted for mean lipid level and lipid-lowering medication and studies without such adjustment for LV with CVDs. There were also some heterogeneities between studies conducted on HICs and studies conducted on UMICs of LV with CVDs and all-cause mortality. An updated analysis should be undertaken when more data become available in the future. In addition, we did not conduct dose–response analysis to quantify the linear or nonlinear relationships between LV and CVDs and all-cause mortality due to the limited data.

The present findings support that LV may be used as an important clinical indicator for the risk of CVDs and all-cause mortality. More sophisticated measures of LV are needed, as well as consensus about how such LV should be defined, including types of lipids, metrics of variability, measurement intervals, times of measurements, and the temporality of the variance [40]. Meanwhile, more high-quality epidemiologic investigations and primary prevention studies are needed to clarify the role of LDL-C variability or TG variability on the risk of CVDs and all-cause mortality.

## 5. Conclusions

In conclusion, lipid variability, particularly the variability in TC and HDL-C, is positively associated with the risk of CVDs and all-cause mortality. Measurement and surveillance of lipid variability beyond mean lipid levels have important clinical implications for the risk assessment of CVDs and all-cause mortality.

## Figures and Tables

**Figure 1 nutrients-14-02450-f001:**
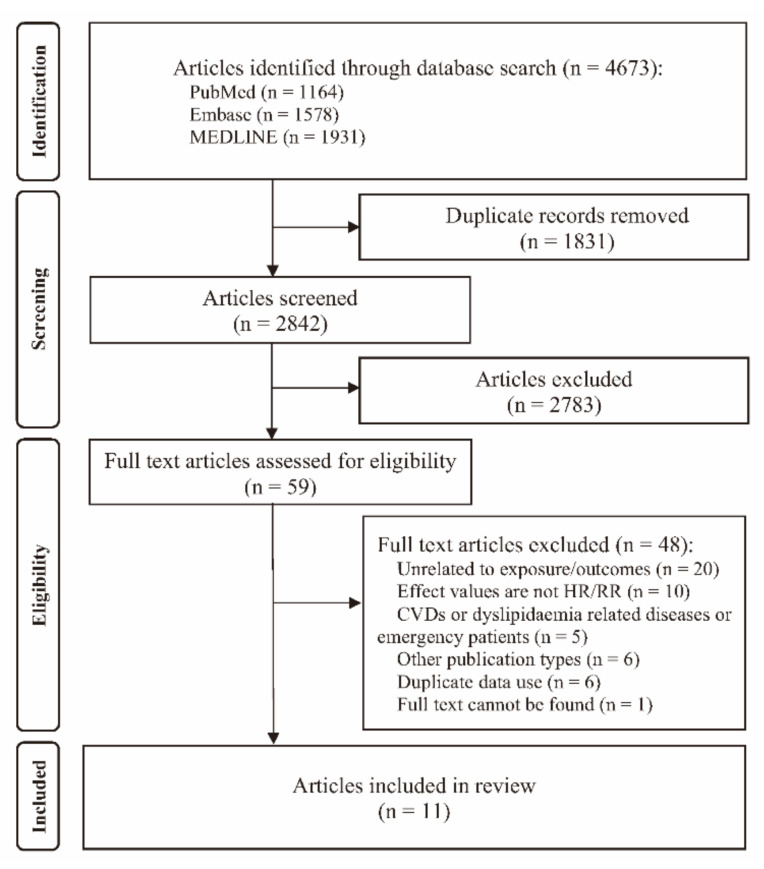
Study screening flowchart.

**Figure 2 nutrients-14-02450-f002:**
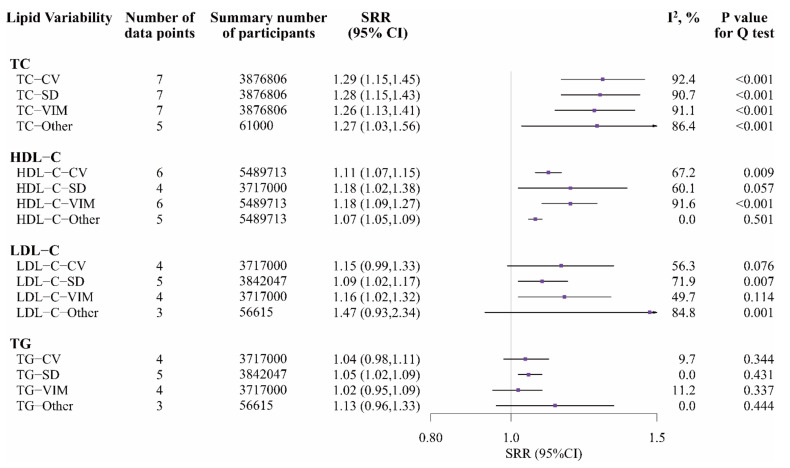
Random-effects meta-analysis of standardized RRs for different types and metrics of LV (top vs. bottom quartile) and CVDs. Notes: If data were reported based on subgroups of CVDs, they were treated as different data points; LV, lipid variability; CVDs, cardiovascular diseases; TC, total cholesterol; HDL-C, high-density lipoprotein cholesterol; LDL-C, low-density lipoprotein cholesterol; TG, triglycerides; CV, coefficient of variation; SD, standard deviation; VIM, variation independent of the mean; TC-Other included average real variability of TC (TC-ARV), standard deviation of the residuals of TC (TC-SDR), and root mean square error of TC (TC-RMSE); HDL-C-Other included average real variability of HDL-C (HDL-C-ARV); LDL-C-Other included average real variability of LDL-C (LDL-C-ARV); TG-Other included average real variability of TG (TG-ARV).

**Figure 3 nutrients-14-02450-f003:**
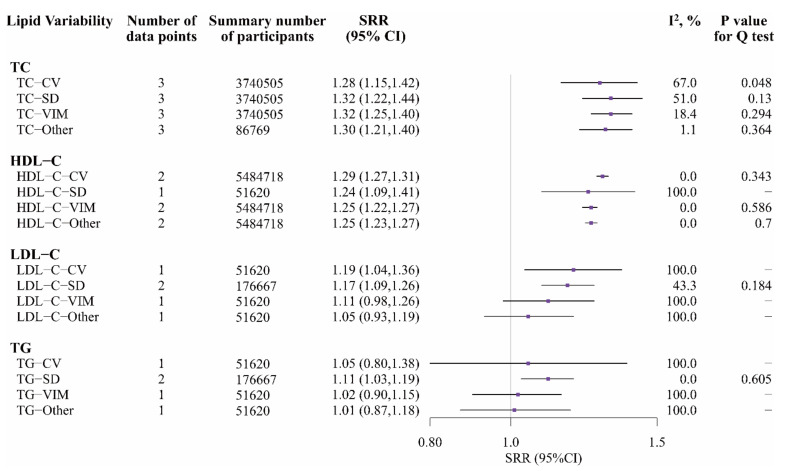
Random-effects meta-analysis of standardized RRs for different types and metrics of LV (top vs. bottom quartile) and all-cause mortality. Notes: If data were reported based on subgroups of CVDs, they were treated as different data points. LV, lipid variability; TC, total cholesterol; HDL-C, high-density lipoprotein cholesterol; LDL-C, low-density lipoprotein cholesterol; TG, triglycerides; CV, coefficient of variation; SD, standard deviation; VIM, variation independent of mean; TC-Other included average real variability of TC (TC-ARV), average successive variability of TC (TC-ASV) and root mean square error of TC (TC-RMSE); HDL-C-Other included average real variability of HDL-C (HDL-C-ARV); LDL-C-Other included average real variability of LDL-C (LDL-C-ARV); TG-Other included average real variability of TG (TG-ARV).

**Table 1 nutrients-14-02450-t001:** Detailed characteristics of the included articles (*n* = 11).

Authors (Year)	Country	Cohort	WB Income Region	Study Period	Mean/Median Follow-Up Years	Number of Participants	Age (Years)	Female (%)	Lipids	Metrics of Variability	Numbers of Causes of Outcome(s)	Comparison
Kreger, et al. (1994) [24]	America	FHS	HICs	1948–1985	NR	2912	30–62	51.7	TC	RMSE	CVDs (CHD); all-cause mortality	Extreme quartiles
Kim, et al. (2017) [14]	South Korea	KNHIS	HICs	2002–2015	8.3	3,656,648	≥20	32.4	TC	CV; SD; VIM	CVDs (stroke/MI); all-cause mortality	Extreme quartiles
Kwon, et al. (2019) [25]	South Korea	KNHIS	HICs	2009–2015	5.3	3,820,191	≥40	47.1	TC	CV; SD; VIM	CVDs (HF)	Extreme quartiles
Zhu, et al. (2019) [26]	China	YHIS	UMICs	2010–2017	4.3	32,237	≥40	NR	TC	CV; SD; VIM; ASV	All-cause mortality	Extreme quartiles
Lee, et al. (2019) [27]	South Korea	KNHIS	HICs	2009–2015	5.4	3,660,385	43.4	31.8	TC; HDL-C; LDL-C; TG	CV; SD; VIM	CVDs (AF)	Extreme quartiles
Liu, et al. (2020) [28]	China	Kailuan cohort	UMICs	2006–2017	7.0	51,620	52.8 ± 11.8	24.0	TC; HDL-C; LDL-C; TG	CV; SD; VIM; ARV	CVDs (MI); all-cause mortality	Extreme quartiles, Per SD
Han, et al. (2020) [6]	South Korea	KNHIS	HICs	2009–2017	5.1	5,433,098	≥20	34.2	HDL-C	CV; VIM; ARV	CVDs (stroke/M); all-cause mortality	Extreme quartiles
Kalani, et al. (2020) [29]	America	The Cardiovascular Health Study	HICs	1989–1998	5.2	1473	73.8 ± 4.4	60.1	TC	SDR	CVDs (stroke)	Per unit
Wang, et al. (2020) [30]	China	Kailuan Cohort	UMICs	2006–2016	6.0	51,620	52.8 ± 11.8	24.0	TC; HDL-C; LDL-C; TG	CV; SD; VIM; ARV	CVDs (stroke)	Extreme quartiles, Per SD
Wan, et al. (2020) [15]	China (Hong Kong)	CDARS	HICs	2008–2017	6.5	125,047	64.3 ± 9.7	54.5	TG; LDL-C	SD	CVDs; all-cause mortality	Extreme quintiles
Huang, et al. (2021) [31]	China	Liaobu Community Study	UMICs	2013–2018	4.2	4995	62.7 ± 12.6	55.2	TC; LDL-C; HDL-C; TG	CV; SD; VIM; ASV	CVDs (stroke)	Extreme quartiles

Notes: FHS, Framingham Heart Study; KNHIS, Korean National Health Insurance System cohort; YHIS, Yinzhou Health Information System; CDARS, The Clinical Data Analysis and Reporting System; WB income region, the World bank income region; HICs, high-income countries; UMICs, upper-middle-income countries; NR, not report; TC, total cholesterol; HDL-C, high-density lipoprotein cholesterol; LDL-C, low-density lipoprotein cholesterol; TG, triglycerides; RMSE, root mean square error; CV, coefficient of variance; SD, standard deviation; VIM, variation independent of the mean; ARV, average real variability; ASV, average successive variability; SDR, standard deviation of the residuals; CVDs: cardiovascular diseases.

**Table 2 nutrients-14-02450-t002:** Summary effects and 95% CI using random-effects subgroup meta-analysis for the associations of TC variability (top vs. bottom quartile) with CVDs.

Characteristics of Studies and Populations	Number of Data Points	SRR (95% CI)	Number of Data Points	SRR (95% CI)	Number of Data Points	SRR (95% CI)
	TC-CV		TC-SD		TC-VIM	
Global analysis	7	1.29 (1.15, 1.45)	7	1.28 (1.15, 1.43)	7	1.26 (1.13, 1.41)
Subtypes of CVDs						
MI	2	1.39 (1.03, 1.87)	2	1.35 (1.03, 1.77)	2	1.39 (1.08, 1.79)
Stroke	3	1.56 (1.07, 2.28)	3	1.59 (1.12, 2.27)	3	1.49 (1.06, 2.10)
AF	1	1.10 (1.06, 1.13)	1	1.09 (1.06, 1.13)	1	1.08 (1.04, 1.12)
HF	1	1.17 (1.13, 1.22)	1	1.18 (1.13, 1.23)	1	1.17 (1.12, 1.22)
Gender *						
Male	4	1.08 (1.05, 1.11)	3	1.09 (1.07, 1.10)	3	1.08 (1.07, 1.10)
Female	4	1.09 (0.99, 1.19)	3	1.06 (1.03, 1.08)	3	1.05 (1.01, 1.09)
Adjusted for mean lipid level						
Yes	6	1.25 (1.12, 1.40)	6	1.24 (1.13, 1.37)	6	1.23 (1.11, 1.36)
No	1	3.83 (2.03, 7.25)	1	4.43 (2.29, 8.56)	1	3.87 (2.04, 7.32)
Adjusted for lipid-lowering medication						
Yes	5	1.43 (1.17, 1.75)	4	1.52 (1.23, 1.86)	4	1.49 (1.23, 1.81)
No	2	1.13 (1.06, 1.21)	3	1.13 (1.07, 1.21)	3	1.12 (1.04, 1.19)

Note: If data were reported based on subgroups of CVDs, they were treated as different data points. The variables used for subgroup meta-analysis included: subtypes of CVDs, gender (male or female), whether adjusting for mean lipid level or not, whether adjusting for lipid-lowering medication or not, whether adjusting for hypertension or not, whether adjusting for diabetes or not, whether adjusting for BMI or not, and whether adjusting for smoking or not: SRR, summary relative risk; CI, confidence interval; CVDs, cardiovascular diseases; TC, total cholesterol; HDL-C, high-density lipoprotein cholesterol; LDL-C, low-density lipoprotein cholesterol; TG, triglycerides; CV, coefficient of variation; SD, standard deviation; VIM, variation independent of the mean; MI, myocardial infarction; AF, atrial fibrillation; HF, heart failure; * three studies explored the relationships between TC-CV, TC-SD, TC-VIM, and CVDs in males and females.

## Data Availability

Not applicable.

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
