# Peer review of "Lipid Variability and Risk of Cardiovascular Diseases and All-Cause Mortality: A Systematic Review and Meta-Analysis of Cohort Studies"

_nutrients, 2022, doi:10.3390/nu14122450_

Round 1

Reviewer 1 Report

The authors conducted a systematic review and meta-analysis of cohort studies to assess and quantify the associations of different lipid variability metrics with CVD and all-cause mortality. They found that lipid variability of total cholesterol, HDL, and coefficient of variance of LDL were positively associated with CVDs and all-cause mortality, whereas variability of TGs had no effect. The authors thus suggest that lipid variability may be useful to optimize assessment for CVD risk and all-cause mortality. Comments follow below.

1) The authors mention the amount of studies used for specific analysis, but not the n for each outcome analyzed. Given the large differences in subjects included in the individual studies (ranging from 1473 to more than 3 million subjects), interpretation of study results may be more informative if the n of subjects would be additionally mentioned in the tables (instead of only informing on number of studies).

2) The term “data point” should be explained better (or reworded) to facilitate understanding of the tables.

3) Interestingly, LV of TC and HDL was predictive for CVD and mortality, but much less so was LDL. While the authors mention potential reasons why robust conclusions were not possible from their analysis, the discussion would benefit from a more extended discussion on why LDL, as the most important predictor of CV risk and as the guideline-recommended target parameter, was less or not predictive. Maybe putting the insignificant results more into perspective of other studies presented in more detail in the discussion may help.

4) Some grammar issues were detected. Proofreading to correct these issues is recommended.

Author Response

Response to Reviewer 1 Comments

The authors conducted a systematic review and meta-analysis of cohort studies to assess and quantify the associations of different lipid variability metrics with CVD and all-cause mortality. They found that lipid variability of total cholesterol, HDL, and coefficient of variance of LDL were positively associated with CVDs and all-cause mortality, whereas variability of TGs had no effect. The authors thus suggest that lipid variability may be useful to optimize assessment for CVD risk and all-cause mortality. Comments follow below.

R1C1: The authors mention the amount of studies used for specific analysis, but not the n for each outcome analyzed. Given the large differences in subjects included in the individual studies (ranging from 1473 to more than 3 million subjects), interpretation of study results may be more informative if the n of subjects would be additionally mentioned in the tables (instead of only informing on number of studies).

We appreciate your instructions, and we have added the summary number of participants for each metric of lipid variability (LV) with cardiovascular diseases (CVDs) in Figure2 and with all-cause mortality in Figure3, as you have suggested.

R1C2: The term “data point” should be explained better (or reworded) to facilitate understanding of the tables.

Thank you very much for pointing this out. As we mention on Page3, Lines144-145 (Main text, Methods): "Any effect sizes stratified by different outcomes are treated as separate data points." If data were reported based on subgroups of CVDs (e.g., stroke and myocardial infarction [MI], separately), they were treated as different data points.

To enable readers to understand better, we have added an explanation of "data point" in the notes to Figure2, Figure3 and Table 2 of the main text.

R1C3: Interestingly, LV of TC and HDL was predictive for CVD and mortality, but much less so was LDL. While the authors mention potential reasons why robust conclusions were not possible from their analysis, the discussion would benefit from a more extended discussion on why LDL, as the most important predictor of CV risk and as the guideline-recommended target parameter, was less or not predictive. Maybe putting the insignificant results more into perspective of other studies presented in more detail in the discussion may help.

Thanks for your constructive suggestions. In terms of the number of studies, cohort studies on the association between LDL-C variability and CVD risk are limited (a total of only four articles with three cohorts on LDL-C-CV in relation to CVDs were included in this systematic review and meta-analysis). And there is some heterogeneity in the identification of outcome events (e.g., stroke, MI), with partially different trends of LV with CVDs between the cohorts, which may have contributed to inconsistent results. Mechanistically, previous studies have found that LDL-C variability may have a greater impact on the occurrence of adverse health events in the group of patients with prior CVDs who are more prone to lipid fluctuations[1-3], whereas TC and HDL-C variability may have a greater impact on the general population[4]. Our study focuses on the population who did not have a history of CVDs. Thus, further studies to examine the mechanisms by which higher cholesterol variability relates to CVDs development, and whether the reduction of cholesterol variability can lower CVDs risk are required, further studies to investigate the relationships of LDL-C and TG variability with CVDs and mortality risks are needed.

We have added such information to the discussion. For ease of review, we are also listing it below:

Pages11-12, Lines325-336 (Main text, Discussion): “Previous studies also have found that LDL-C variability may have a greater impact on the occurrence of adverse health events in the group of patients with prior CVDs who are more prone to lipid fluctuations[13,38,39]. Nevertheless, the capacity of limited number of studies with different exposure-outcome combinations may not allow us to draw robust conclusions regarding the insignificance of LDL-C variability and TG variability with CVDs and all-cause mortality of general population in the sensitivity analysis. Thus, further studies to investigate the relationships of LDL-C and TG variability with CVDs and mortality risks are needed, further studies to examine the mechanisms by which higher cholesterol variability relates to CVDs development, and whether the reduction of cholesterol variability can lower CVDs risk are required. We call for more researchers from different countries and regions to focus on different types and metrics of LV and adverse health outcomes.”

R1C4: Some grammar issues were detected. Proofreading to correct these issues is recommended.

Thank you for pointing this out. We have carefully checked all spelling and grammar throughout the manuscript and supplementary file.

Reference:

  1. Bangalore, S.; Breazna, A.; DeMicco, D.A.; Wun, C.C.; Messerli, F.H. Visit-to-visit low-density lipoprotein cholesterol variability and risk of cardiovascular outcomes: insights from the TNT trial. Journal of the American College of Cardiology 2015, 65, 1539-1548.
  2. Boey, E.; Gay, G.M.; Poh, K.K.; Yeo, T.C.; Tan, H.C.; Lee, C.H. Visit-to-visit variability in LDL- and HDL-cholesterol is associated with adverse events after ST-segment elevation myocardial infarction: A 5-year follow-up study. Atherosclerosis 2016, 244, 86-92.
  3. Bangalore, S.; Fayyad, R.; Messerli, F.H.; Laskey, R.; DeMicco, D.A.; Kastelein, J.J.; Waters, D.D. Relation of Variability of Low-Density Lipoprotein Cholesterol and Blood Pressure to Events in Patients With Previous Myocardial Infarction from the IDEAL Trial. The American journal of cardiology 2017, 119, 379-387.
  4. Wang, A.; Li, H.; Yuan, J.; Zuo, Y.; Zhang, Y.; Chen, S.; Wu, S.; Wang, Y. Visit-to-Visit Variability of Lipids Measurements and the Risk of Stroke and Stroke Types: A Prospective Cohort Study. Journal of stroke 2020, 22, 119-129.

Reviewer 2 Report

In this review, the authors conducted a meta-analysis of published studies on cardiovascular risk and lipid variability assessed by different metrics. Data collection and subsequent statistical analysis were very thorough, although the heterogeneity of the studies analyzed did not allow for many parameters to establish a definite correlation with future cardiovascular events. The exposition of the results in my opinion is too long and at times tedious distracting the reader from the essential messages; it should be shortened by relying on additional tables to represent the statistical significance for each variable analyzed and leaving in the text the exposition of only the data essential for comprehension.

It would be helpful to enclose a summary table of all the many acronyms that appear in the text

Specific comments.

Paragrah abstract. Lines 43-45. The meaning of the sentence “The results of subgroup and meta-regression showed that with and without adjustment for lipid-lowering medications and mean lipid levels were significant covariates influencing the heterogeneity between studies” is not clear to me.

Paragraph discussion. Line 409. The acronym RSME appears for the first time and should be spelled out in full

Author Response

Response to Reviewer 2 Comments

In this review, the authors conducted a meta-analysis of published studies on cardiovascular risk and lipid variability assessed by different metrics. Data collection and subsequent statistical analysis were very thorough, although the heterogeneity of the studies analyzed did not allow for many parameters to establish a definite correlation with future cardiovascular events.

Thank you for your supportive comments. We believe that our manuscript has greatly benefited from your and other reviewers’ comments.

R2C1: The exposition of the results in my opinion is too long and at times tedious distracting the reader from the essential messages; it should be shortened by relying on additional tables to represent the statistical significance for each variable analyzed and leaving in the text the exposition of only the data essential for comprehension.

Thanks for your constructive suggestions. We have revised the relevant statements to the presentation of the results in the abstract and main text to make them more unambiguous, hoping that these amendments can provide readers with more accurate information.

For ease of review, we list relevant modifications in page1, lines 43-45(Abstract), page8, lines186-187, 204-208 (Main text, Methods), page9, lines220-221, 231-240 (Main text, Results), page10, lines251-270 (Main text, Results) and page11 lines 325-328(Main text, Discussion), page12, lines 332-336.

R2C2: It would be helpful to enclose a summary table of all the many acronyms that appear in the text.

Thanks for your suggestion. We have enclosed a summary table of all acronyms that appear in the text and put it at the end of the main text, pages13-14, hoping that these amendments can increase readability for readers.

R2C3: Paragrah abstract. Lines 43-45. The meaning of the sentence “The results of subgroup and meta-regression showed that with and without adjustment for lipid-lowering medications and mean lipid levels were significant covariates influencing the heterogeneity between studies” is not clear to me.

Thank you very much for pointing this out. We apologise for any confusion that may have arisen due to our misrepresentation. In fact, we were trying to say that our study found that adjustment for lipid-lowering medication or not, adjustment for mean lipid level or not might be sources of heterogeneity between studies of LV with CVDs.

In order to make this statement more accurate, we have revised the corresponding section as follows:

Page1, Lines43-45 (Abstract): "The effects of SRR became stronger when analyses were restricted to studies that adjusted for lipid-lowering medication and unadjusted for mean lipid levels."

R2C4: Paragraph discussion. Line 409. The acronym RSME appears for the first time and should be spelled out in full.

Thank you for pointing this out. We deeply regret the confusion caused by our typo of the acronym RSME. Actually, the correct spelling is RMSE (root mean square error), as you can see in Page3, Line119 (Main text, Methods). For ease of review, we have revised relevant contents as follows:

Page12, Line363 (Main text, Discussion): “However, several limitations should be considered. First, due to the limited numbers of available studies, the LV effects of the ARV, ASV, RMSE, and SDR metrics could not be taken into account separately.”

Meanwhile, we have carefully checked all spelling and grammar throughout the manuscript and supplementary file.

Reviewer 3 Report

This review entitled “Lipid variability and risk of cardiovascular diseases and all-2 cause mortality: a systematic review and meta-analysis of cohort studies conducted by Shuting Li et al. is an interesting study that shows that there is an association between variability in lipid determination with cardiovascular disease and mortality. The measurement of lipid variability with suitable instruments can have important clinical implications.

The limitations of the study regarding the linearity of causality are also described.

Author Response

Response to Reviewer 3 Comments

This review entitled “Lipid variability and risk of cardiovascular diseases and all-cause mortality: a systematic review and meta-analysis of cohort studies conducted by Shuting Li et al. is an interesting study that shows that there is an association between variability in lipid determination with cardiovascular disease and mortality. The measurement of lipid variability with suitable instruments can have important clinical implications.

The limitations of the study regarding the linearity of causality are also described.

Thank you for your supportive comments.

Round 2

Reviewer 2 Report

The changes made are satisfactory.

The current version of this paper allows for smoother reading and better understanding of the results